# Functional Analysis of *PdbERF109* Gene Regulation of Salt Tolerance in *Populus davidiana* × *P. bolleana*

**DOI:** 10.3390/plants14172800

**Published:** 2025-09-06

**Authors:** Nan Jiang, Shixian Liao, Ruiqi Wang, Wenjing Yao, Yuting Wang, Guanzheng Qu, Tingbo Jiang

**Affiliations:** 1State Key Laboratory of Tree Genetics and Breeding, Northeast Forestry University, Harbin 150040, China; nikkijiang23@163.com (N.J.); lsx1782801124@163.com (S.L.); 870831997@nefu.edu.cn (R.W.); yaowenjing@njfu.edu.cn (W.Y.); wyt1513026188@163.com (Y.W.); 2Co-Innovation Center for Sustainable Forestry in Southern China, Nanjing Forestry University, 159 Longpan Road, Nanjing 210037, China

**Keywords:** *Populus davidiana* × *P. bolleana*, ERF transcription factor, salt stress, ROS scavenging, transcription regulation

## Abstract

ERF family transcription factors are crucial regulators in plants, playing a central role in abiotic stress responses and serving as important targets for stress-tolerant crop breeding. *Populus davidiana* × *P. bolleana*, an elite hybrid poplar cultivar artificially selected in northern China, holds significant research value encompassing ecological restoration, economic industries, genetic resource development, and environmental adaptability. This study identified that *PdbERF109* expression was significantly upregulated in *P. davidiana* × *P. bolleana* response to salt treatment. Furthermore, transgenic poplar lines overexpressing *PdbERF109* (*OE*) were generated. Salt stress assays demonstrated that *PdbERF109* overexpression significantly enhanced salt tolerance in transgenic poplar. Compared to wild-type (WT) plants, *PdbERF109*-OE lines exhibited a significant enhancement in the activities of antioxidant enzymes, with increases of 2.3-fold, 1.2-fold, and 0.5-fold for superoxide dismutase (SOD), peroxidase (POD), and catalase (CAT), respectively, while the levels of malondialdehyde (MDA) and hydrogen peroxide (H_2_O_2_) were markedly reduced by 39.89% and 40.03%, indicating significantly enhanced reactive oxygen species (ROS) scavenging capacity and reduced oxidative damage. Concurrently, *PdbERF109* overexpression reduced the natural leaf relative water loss (%). Meanwhile, yeast one-hybrid assays confirmed that the *PdbERF109* protein specifically binds to GCC-box and DRE cis-acting elements. This study established *PdbERF109* as a positive regulator of salt stress responses, highlighting its potential as a target gene for improving plant tolerance to high salinity, providing a promising candidate gene for the molecular breeding of salt-tolerant crops.

## 1. Introduction

Soil salinization refers to the process of environmental degradation in soil resulting from the accumulation of soluble salts in the surface layer. Global-scale studies indicate that saline soils are distributed across all climatic zones and continents, with an estimated global area ranging from 8.31 to 11.73 million square kilometers (Mkm^2^) [1,2]. Soil salinization-alkalization is a major environmental stressor that severely impairs plant growth. To cope with salt stress, plants have evolved multifaceted adaptive mechanisms involving physiological, biochemical, and molecular responses. These include regulating ion and water homeostasis, maintaining photosynthetic function, and mitigating oxidative damage [3]. In recent years, the molecular mechanisms underlying plant responses to salt stress have become a major research focus in China. Under stress conditions, plants are able to perceive environmental signals and modulate the transmission of various signaling pathways, thereby regulating the expression of stress-responsive genes and adjusting to external stimuli [4]. Gene editing technologies now allow for precise dissection of these genes and their functions, with evidence showing that targeted editing can enhance plant resilience under adverse conditions [5]. In China, the reclamation and utilization of saline-alkali land are critical for ecological security and sustainable agriculture. The molecular breeding of salt-tolerant crops thus offers a promising strategy for the improved use of these marginal soils.

The redox enzyme system is one of the earliest defense mechanisms activated in plants upon exposure to environmental stress. When plants are subjected to stress induced damage, they rapidly accumulate reactive oxygen species (ROS), which act as signaling molecules to trigger defense responses. However, excessive accumulation of ROS, particularly H_2_O_2_, can cause oxidative damage to plant cells. Elevated ROS levels and free radicals may disrupt cellular membrane integrity by promoting lipid peroxidation, ultimately leading to malondialdehyde (MDA) accumulation and cellular injury or death. To counteract this, plants activate enzymatic antioxidant systems, including catalase (CAT), peroxidase (POD), and superoxide dismutase (SOD), maintaining redox balance by donating electrons to ROS. This electron transfer mechanism both directly neutralizes existing ROS and interrupts their chain reactions, thereby effectively suppressing excessive ROS accumulation and preventing oxidative damage to cellular structures and biomacromolecules [6]. Existing studies confirm that during initial stress exposure, the activity of these three enzymes in leaves shows a significant upregulation [7]. When *Jatropha curcas* L. is treated with salt, the activity of SOD, POD, and CAT in plants gradually increases as the salt concentration increases during the initial stage of stress [8]. At the molecular level, the overexpression of TFs that regulate plant salt tolerance significantly enhances adaptive capacity. This process induces the synergistic upregulation of antioxidant enzymes, effectively inhibiting the accumulation of H_2_O_2_ and MDA, preserving membrane system functionality, and ultimately conferring stronger salt adaptation. Studies demonstrate that in *Arabidopsis thaliana* overexpressing *MbERF12* under salt stress exhibits elevated activities of POD, CAT, and SOD, reduced MDA levels, and enhanced ROS scavenging capacity [9]. Under salt stress, transgenic *SmERF1* of *Salvia miltiorrhiza* tobacco lines exhibited enhanced salt tolerance. These transgenic lines elevated activities of SOD and POD compared to wild-type (WT) plants, while MDA content was markedly lower [10]. Consequently, to advance our understanding of salt tolerance mechanisms in woody plants and to breed elite salt-tolerant cultivars, further investigation is required to determine whether other transcription factors (TFs) possessing salt tolerance functions can mitigate plant damage under salt stress by scavenging ROS.

Transcription factors play irreplaceable roles as key regulators in plant responses to abiotic stress, being deeply involved in signal transduction, physiological regulation, and functional adaptation processes. Among them, AP2/ERF TFs contribute significantly to plant growth and development, as well as to the regulation of responses to various abiotic stresses [11,12]. For instance, in *Triticum aestivum*, transgenic seedlings overexpressing the *TaERF3* gene exhibited significantly enhanced tolerance to salt stress. Overexpressed *TaERF3* lines showed significantly increased accumulation levels of proline and chlorophyll in leaves, while H_2_O_2_ content and stomatal conductance significantly decreased [13]. In *Oryza sativa*, the overexpression of *OsERF101* increased proline content and peroxidase activity, consequently enhancing drought tolerance in transgenic plants. Conversely, knockdown lines exhibited impaired drought tolerance [14]. In *Vitis vinifera* L., the overexpression of *VvERF63* contributes to cold tolerance by enhancing photosynthetic capacity and alleviating cellular damage. Reciprocally, *VvERF63* knockdown plants incurred significant deficits in cold acclimation ability [15].

Research indicates that ERF family members help plants mitigate the adverse effects of high salinity. In *Solanum tuberosum*, silencing the *StERF3* gene enhanced tolerance to NaCl stress, concomitant with the activation of defense-related genes such as *PR1*, *NPR1*, and *WRKY1*. Conversely, the overexpression of *StERF3* resulted in a reduced expression of these defense-related genes [16]. In *Brassica oleracea* var. *italica*, transgenic plants overexpressing *BoERF1* displayed higher seed germination rates and reduced chlorophyll loss under salt stress, indicating that *BoERF1* acts as a positive regulator of salt tolerance [17]. In *Tamarix chinensis*, *ThCRF1* responds to salt stress and can bind DRE and GCC-box to regulate pathogen and abiotic stress-related genes. Transient transformation experiments producing *ThCRF1* overexpression and *ThCRF1*-RNAi-silenced plants revealed that *ThCRF1* overexpression significantly increased tolerance to salt stress, while RNAi silencing of *ThCRF1* significantly decreased tolerance [18]. In *Oryza sativa*, functional analysis revealed that transgenic lines overexpressing *OsERF922* exhibited reduced tolerance to salt stress in contrast to RNAi-silenced lines, which showed enhanced salt tolerance [19]. These studies enrich our understanding of the molecular mechanisms by which ERF transcription factors regulate plant responses to salt stress. However, most of the functional studies of AP2/ERF family members are currently focused on herbaceous plants, and there are fewer reports on the function of ERF salinity tolerance in woody plants or trees.

*Populus davidiana* × *P. bolleana* is a widely cultivated hybrid poplar in northern China with significant value for environmental remediation and economic benefits. The aim of this study was to functionally characterize the role of *PdbERF109* (Potri.009G101900) from *P. davidiana* × *P. bolleana* in salt stress tolerance. Specifically, we sought to determine whether *PdbERF109* confers salt resistance by enhancing antioxidant defense mechanisms and reducing oxidative damage. To achieve this, we quantified the expression levels of 15 *PdbERF* genes in response to salt treatment, based on the gene structure of ERF family members in *Populus trichocarpa* [20]. We then generated *PdbERF109* overexpressing transgenic poplars, characterized their phenotypic responses under salt stress, and analyzed key physiological and biochemical indicators of oxidative stress. This study provides crucial functional evidence for *PdbERF109* as a positive regulator of salinity tolerance and offers a validated genetic resource for breeding salt-resistant poplar varieties.

## 2. Materials and Methods

### 2.1. Plant Material

*P. davidiana* × *P. bolleana* seedlings were propagated on half-strength Murashige and Skoog (1/2MS) medium supplemented with 0.1 mg·L^−1^ indole-3-butyric acid (IBA) and 0.01 mg·L^−1^ α-naphthaleneacetic acid (NAA) with pH 5.8. Tissue-cultured seedlings were maintained under a 14 h light/10 h dark photoperiod at 22 ± 1 °C, with a photosynthetic photon flux density of 150 μmol m^−2^ s^−1^ and relative humidity of 50–60%. Three-week-old tissue culture seedlings were transplanted into soil culture. The soil culture environment consisted of a substrate mixed with black soil and vermiculite (3:1, *v*/*v*), with a temperature of 25 °C and air humidity of 50–60%. Healthy seedlings grown in soil for 45–60 days will undergo stress treatment.

For the stress treatment of wild-type plants, 150 mM NaCl was used to treat *P. davidiana* × *P. bolleana* tissue-cultured seedlings. Three biological replicates were sampled at 0, 3, 6, 12, 24, and 48 h post-treatment. The roots, stems, and leaves of each seedling were collected.

To determine salt tolerance in the overexpressed poplar, three tissue-cultured poplar seedlings overexpressing *PdbERF109* and two WT seedlings were transplanted into a black soil/vermiculite (3:1, *v*/*v*) substrate. Following 45–60 days of greenhouse acclimatization, plants exhibiting comparable growth vigor were subjected to cyclic salt stress (100 mL of 150 mM NaCl per application every 3 days for 14 days), with equivalent volumes of water counterparts serving as controls. Growth phenotypes were documented after the treatment period. All primers used in this study (Appendix A) were synthesized by Sangon Biotech.

### 2.2. Screening and Cloning of PdbERF109

Total RNA was extracted using the CTAB method [21], reverse transcribed into cDNA using a commercial kit (StarScript Pro All-in-one RT Mix with gDNA Remover, GenStar, Suzhou, China).

The primers (A19-F/A19-R) were designed based on the *P. trichocarpa* transcript sequence (Phytozome ID: Potri.009G101900), with promoter-specific primers (proA19-F/proA19-R) targeting a 2000 bp region upstream of the coding sequence start codon. The full-length coding sequence was amplified from poplar cDNA using KOD HiFi DNA Polymerase, while the promoter fragment was amplified from genomic DNA using the same polymerase.

Gel-purified amplicons were A-tailed with rTaq DNA Polymerase, then cloned into the pMD19-T vector and transformed into DH5α competent cells. Positive colonies were sequenced after overnight incubation. Validated clones showing 100% sequence identity to the reference were resuspended in 50% (*v*/*v*) glycerol (600 μL bacterial culture + 600 μL glycerol) for cryopreservation at −80 °C.

### 2.3. Bioinformatics Analysis of PdbERF109

The conserved domain of *PdbERF109* was identified using NCBI (https://www.ncbi.nlm.nih.gov/Structure/cdd/wrpsb.cgi, accessed on 19 April 2025), the signal peptide was predicted using (https://services.healthtech.dtu.dk/services/SignalP-5.0/, accessed on 19 April 2025), the secondary structure was analyzed using (https://npsa-prabi.ibcp.fr/cgi-bin/npsa_automat.pl?page=npsa_sopma.html, accessed on 19 April 2025), the tertiary structure was constructed using (https://swissmodel.expasy.org/interactive, accessed on 20 April 2025), hydrophilicity and hydrophobicity were predicted using Expasy (https://web.expasy.org/protscale/, accessed on 20 April 2025), and the transmembrane structure was predicted using TMHMM 2.0 (https://services.healthtech.dtu.dk/services/TMHMM-2.0/, accessed on 21 April 2025). The downstream target gene was determined using PlantTFDB (https://plantregmap.gao-lab.org/network.php, accessed on 1 July 2025). The components contained in the *PdbERF109* promoter sequence were analyzed by (https://bioinformatics.psb.ugent.be/webtools/plantcare/html/, accessed on 5 July 2025) [22].

### 2.4. Quantitative Real-Time PCR Analysis

To comprehensively characterize the spatiotemporal dynamics of *PdbERF109* expression in poplars under salt stress, the tissue-specific and time-dependent expression patterns were further analyzed. Total RNA was extracted using the CTAB method and reverse-transcribed into cDNA using a commercial kit (StarScript Pro All-in-one RT Mix with gDNA Remover, GenStar, Suzhou, China) for qRT-PCR amplification using fluorescent quantitative PCR instrument (qTOWER3G, Germany). Gene-specific primers for *PdbERF109* employed A19-qPCR-F/A19-qPCR-R and reference gene primers employed Act-F/Act-R(Actin gene ID: JM986590). Relative expression levels were calculated using the 2^−△△Ct^ method.

### 2.5. Subcellular Localization and Transcriptional Activation Activity

The full-length *PdbERF109* coding sequence (stop codon excluded) was amplified from pMD19-T-*PdbERF109* plasmid DNA using primers A19-XbaI-F/A19-SpeI-R containing XbaI/SpeI restriction sites. The digested fragment was ligated into the pBI121-GFP vector, generating the plant localization construct pBI121-*PdbERF109*-GFP. Both the fusion construct and empty pBI121 vector were separately transformed into GV3101 competent cells. Transient transformation of *Nicotiana benthamiana* leaves was performed via syringe infiltration. After 48 h dark incubation, GFP fluorescence was visualized using a Zeiss LSM 800 laser scanning confocal microscope (Carl Zeiss AG, Oberkochen, Germany) 236 to determine *PdbERF109* subcellular localization. The full-length *PdbERF109* was cloned into pGBKT7 via homologous recombination to generate pGBKT7-*PdbERF109*. This construct, alongside empty pGBKT7 (negative control) and pGBKT7-53 (positive control), was transformed into Y2H Gold yeast competent cells. Transformants were selected on SD/-Trp medium, then streaked onto SD/-Trp/-His/X-α-Gal medium. Transcriptional activation activity was confirmed by blue colony development after 3-day incubation at 30 °C.

### 2.6. Generation and Validation of Transgenic Poplar

Transgenic *P. davidiana* × *P. bolleana* lines were generated via *Agrobacterium* strain GV3101-mediated leaf disk transformation using the fusion vector pBI121-*PdbERF109*-GFP [23]. Leaves (3rd–5th positions) from 4- to 6-week-old tissue-cultured seedlings were excised along the midveins for infection. Kanamycin (Kan)-resistant shoots were induced on selection medium (25 mg·L^−1^ Kan), followed by root induction on equivalent Kan-containing medium. Putative transgenic lines were validated by PCR amplification of genomic DNA using GFP-specific primers with plasmid DNA as a positive control and WT DNA as a negative control. *PdbERF109* expression levels in transgenic lines were quantified by qRT-PCR.

### 2.7. Leaf Relative Water Loss (%)

Leaf relative water loss (%) was determined using the second to sixth leaves collected from WT and transgenic poplars, and five leaves per plant were pooled as a single sample unit. Fresh weight (FW) was immediately recorded after collection. Subsequently, the collected leaves were weighed every 0.5 h (TW) under controlled conditions (25 °C) over an 8 h period. Final dry weight (DW) was obtained after oven-drying leaves at 80 °C overnight. The relative water loss (%) was calculated using the following formula: Relative Water Loss (%) = [(FW − TW)/(FW − DW)] × 100, with three biological replicates per experimental group.

### 2.8. Measurement of Physiological Parameters and Histochemical Staining

For the physiological assays, leaves were collected after NaCl stress to quantify the contents of MDA and H_2_O_2_, as well as the activities of SOD, CAT, and POD using commercial kits (Beijing Boxbio Science & Technology Co., Ltd., Beijing, China). There were three biological replicates per line.

For histochemistry, leaves were stained with nitro blue tetrazolium chloride (NBT)and 3,3′-diaminobenzidine (DAB) [24], then completely decolorized in 95% ethanol and photographed against a white background, with three technical replicates.

### 2.9. Yeast Hybrid Assays

To determine whether *PdbERF109* specifically interacts with the GCC-box element (GCCGCC) and DRE element (ACCGAC), the core sequences of both elements were triplicated as tandem repeats and cloned into the pHIS2 vector to drive HIS reporter expression. Full-length *PdbERF109* cDNA was ligated into pGADT7-Rec2, generating effector construct pGADT7-Rec2-*PdbERF109*. Recombinant effectors were co-transformed with element-pHIS2 reporters into Y187 yeast competent cells. After transformation, positive colonies were selected on SD/-Trp/-His/-Leu medium (TDO) and validated through serial dilutions (10×, 100×, 1000×) on TDO containing 50 mM 3-amino-1,2,4-triazole (3-AT). Controls included positive (p53HIS2 + pGADT7-Rec2-53, p53 AD binds its cognate site, activating HIS3 expression) and negative (p53HIS2 + pGADT7-Rec2-*PdbERF109*, *PdbERF109* cannot bind p53 sites, hence no HIS3 activation).

### 2.10. Data Analysis

Standard errors and standard deviations were calculated by using SPSS 21 (Chicago, IL, USA). A statistically significant level was set to a *p*-value  ≤  0.05. Comparative analyses of the two sets of data were performed using two-way ANOVA. The data are presented as mean ± standard error (SE), with each SE being calculated from three independent biological samples.

## 3. Results

### 3.1. Expression Profiling of PdbERF Genes Under Salt Stress

To identify salt-responsive candidate regulators within 15 *PdbERF* genes, qRT-PCR profiling in root tissue analysis was performed to assess temporal expression patterns. The results demonstrate that under salt stress for 24 h, 14 genes showed significantly increased expression levels, 1 gene showed decreased expression, and 3 of these genes showed expression levels increased by more than 150-fold. Compared to basal levels at 0 h, *PdbERF109* transcript abundance increased by 172.6-fold after 24 h of salt treatment. According to the expression amount and expression trend, *PdbERF109* exhibited significant expression fluctuations relative to other genes in *P. davidiana* × *P. bolleana* response to salt stress (Figure 1a).

qRT-PCR profiling revealed tissue-specific expression patterns of *PdbERF109* in poplar, with the highest transcript levels in leaves, followed by the stems and roots. Under salt stress, *PdbERF109* exhibited pronounced early upregulation in roots or leaves, while expression varied significantly over time without notable changes in stems. In leaves, significant differences in *PdbERF109* transcript levels were observed at 12 h, 24 h, and 48 h, decreasing by 49.2%, 45.61%, and 45.6%, respectively, compared to untreated controls. In contrast, transcript levels in root tissues reached their peak at 24 h, exhibiting a 172.6-fold increase relative to untreated controls. At 3 h, it was 98.1-fold, at 6 h, it was 155.8-fold, at 12 h, it was 63.5-fold, and at 48 h, it was 81.6-fold (Figure 1b).

### 3.2. Bioinformatic Characterization of PdbERF109

To elucidate the biological characteristics of *PdbERF109*, we summarize the following research results: the CDS of *PdbERF109* spans 801 bp, encoding a 267-amino acid protein containing a conserved AP2/ERF domain (Figure 2a). Bioinformatic analysis predicted that *PdbERF109* has a molecular weight of 29,486.25 Da, a theoretical pI of 8.52, and a formula of C_1281_H_2019_N_369_O_401_S_15_. Meanwhile, the instability index and grand average hydropathy were 47.38 and 0.727, respectively, indicating that *PdbERF109* belongs to the instability and hydrophilicity protein (Figure 2b). Structural predictions confirmed that the *PdbERF109* protein lacks transmembrane domains and signal peptides. The secondary structure comprises 23.68% α-helices, 2.26% extended strands, and 74.06% random coils (Figure 2c), with the tertiary structure modeled via homology-based prediction (Figure 2d).

### 3.3. Promoter Analysis and Transcription Factor Prediction of PdbERF109

To investigate the possible involvement of *PdbERF109* in biological pathways, a 1795 bp promoter region upstream of *PdbERF109* (*PdbERF109-Pro*) was cloned. An analysis of *PdbERF109-Pro* using PlantCARE revealed multiple stress-responsive cis-elements, including ABRE (ACGT), MYC (CACATG), MYB (TAACCA), STRE (AGGGG), and ARE (TGAC) (Figure 3a).

### 3.4. PdbERF109 Is a Nuclear-Localized Protein with Transcriptional Activation Activity

To determine PdbERF109 subcellular localization, the fusion construct pBI121-*PdbERF109*-GFP and control *35S-GFP* were transiently expressed in *Nicotiana benthamiana* epidermal cells. Fluorescence microscopy revealed pan-cellular GFP signal distribution for the control, while pBI121-*PdbERF109*-GFP localized exclusively to nuclei, confirming *PdbERF109* as a nuclear protein (Figure 3b).

Transcriptional activation was assessed by transforming pGBKT7-*PdbERF109*, the negative control, and positive control, respectively, into Y2H yeast. The result showed that yeast harboring the negative control has no growth, while both the positive control and pGBKT7-*PdbERF109* strains developed blue colonies on selective SD/-Trp/-His/X-α-Gal medium, demonstrating that PdbERF109 has transcriptional activation activity (Figure 3b). To identify the activation domain, progressively truncated PdbERF109 fragments (dC1-dC9) were ligated with pGBKT7 and transformed into Y2H yeast. The minimal fragment with transcriptional activation activity was dC7 (222–248 aa) (Figure 3c).

### 3.5. Molecular Validation of PdbERF109-Overexpressing Transgenic Poplar

To investigate the function of *PdbERF109*, we constructed *PdbERF109*-overexpressing lines (OE) via *Agrobacterium tumefaciens*-mediated leaf disk transformation. PCR screening confirmed five independent transgenic lines. When cultured on rooting medium containing 50 mg/L Kan, transgenic poplars developed normal roots, whereas WT exhibited complete rooting inhibition (Figure 4a). Additionally, qRT-PCR analysis revealed that *PdbERF109* transcript levels were significantly increased in all five transgenic lines compared to the WT (Figure 4b). The three highest-expressed lines were selected for subsequent salt tolerance analyses.

### 3.6. Phenotypic Analysis of PdbERF109-Overexpressing Poplar

To clarify the function of *PdbERF109* in plant growth and development, we observed that *PdbERF109*-overexpressing lines exhibited significantly reduced growth compared to WT under normal conditions. The OE1, OE2, and OE3 lines showed height reductions of 23.4%, 39.8%, and 40%, respectively; root length shortening by 27.1%, 49.2%, and 61.0%; shoot fresh weight decreases of 44.3%, 64.0%, and 52.7%; and root fresh weight reductions of 53.4%, 75.1%, and 85.8% relative to the WT (Figure 5a,b).

### 3.7. Leaf Relative Water Loss (%) Analysis

To assess water retention capacity, a key factor influencing salt tolerance, leaves (second to sixth) from WT and transgenic poplar were monitored under room temperature conditions. Both WT and transgenic poplar leaves exhibited progressively increasing relative water loss (%) over time. Notably, transgenic lines showed significantly reduced water loss compared to WT from 4.5 to 8 h (Figure 6a). The result demonstrated that *PdbERF109* overexpression enhances water retention capacity in transgenic poplar.

### 3.8. Enhanced Salt Tolerance in PdbERF109-Overexpressing Poplar

Under salt stress, WT poplars exhibited earlier wilting than ERF109-overexpressing transgenic poplars OE1, OE2, and OE3. By day 14, all the leaves of WT poplars had died, whereas only partial leaf wilting occurred in transgenic OE lines (Figure 6b). This indicated that *PdbERF109* overexpression significantly enhanced the survival rate of *P. davidiana* × *P. bolleana* under salt stress. Plants produce ROS to induce stress responses under adverse conditions. However, balancing excess ROS accumulation protects plants from oxidative damage. In this study, the activities of SOD, POD, and CAT, as well as the contents of MDA and H_2_O_2_, were measured in the leaves of WT and transgenic OE poplars under both water and salt stress conditions. The result showed that no significant differences were observed in SOD, POD, and CAT activities or in MDA and H_2_O_2_ contents between WT and OE lines under water conditions. Compared to the WT, the average activities of SOD, POD, and CAT in the transgenic OE lines were significantly increased by 2.33-, 1.24-, and 0.52-fold, respectively, under salt stress. Conversely, MDA and H_2_O_2_ contents decreased by 39.89% and 40.03% in the OE lines (Figure 6d). These findings demonstrate that *PdbERF109* overexpression enhances ROS scavenging capacity, consequently reducing H_2_O_2_ accumulation and alleviating plant cell membrane damage. This conclusion was further supported by histochemical staining with NBT and DAB. The staining intensity showed no significant difference between WT and transgenic plants in the water treatment group. Under salt stress, the OE lines displayed markedly less staining than the WT, indicating substantially lower O_2_^−^ and H_2_O_2_ accumulation in transgenic leaves. In conclusion, *PdbERF109* overexpression enhances ROS scavenging capacity to improve salt tolerance in poplar (Figure 6c).

### 3.9. Analysis of PdbERF109-Binding Cis-Acting Elements

During the past two decades, it has been reported that many cis-acting elements can be involved in plant growth and development and exert a response to abiotic stress, and TFs from the same family may specifically recognize and bind specific elements. It has been documented that AP2/ERF transcription factors can specifically bind to elements such as GCC and DRE and participate in abiotic stress responses [25]. To investigate the DNA-binding specificity of *PdbERF109*, the GCC-box and DRE element were each tandemly repeated three times. These repeats were subsequently fused to the reporter vector pHIS2 and co-transformed with the effector vector pGADT7-Rec2-*PdbERF109* into Y187 yeast cells. Yeast transformants were serially diluted and plated onto TDO and TDO-50 mM 3-AT medium. All yeast strains, including controls and test combinations, grew normally on TDO medium. On TDO-50 mM 3-AT medium, yeast cells harboring a positive plasmid exhibited normal growth. Crucially, yeast cells co-expressing pGADT7-Rec2-*PdbERF109* and either the GCC or DRE cis-element also displayed normal growth. In contrast, yeast cells containing a negative plasmid failed to grow on TDO-50 mM 3-AT medium. These results demonstrate that *PdbERF109* specifically binds both GCC and DRE cis-elements (Figure 7).

## 4. Discussion

ERF TFs play pivotal regulatory roles in plant growth, development, and stress responses. They orchestrate physiological processes across distinct developmental stages, modulating morphogenetic progression. Meanwhile, ERF TFs are integral to the defense responses and regulatory functions activated during plant adaptation to abiotic and biotic stresses [26]. The functional divergence observed among orthologous ERF TFs across species enriches our multidimensional understanding of their biological roles while providing critical insights into evolutionary patterns and mechanisms. In this study, qRT-PCR analysis confirmed that *PdbERF109* expression was significantly upregulated in *P. davidiana* × *P. bolleana* under salt stress, suggesting its potential involvement in physiological response to salinity. Currently, studies have shown that homologues of *PdbERF109* are involved in stress resistance in other plants. In *A. thaliana*, the overexpression of ethylene response factor 109, *ERF109,* can enhance salt tolerance [27]. In *Brassica rapa* subsp. *pekinensis*, *BrERF109* acts as a positive regulator; its overexpression reduces ABA sensitivity, enhances antioxidant enzyme activities and chlorophyll retention, and induces canonical stress-responsive gene expression, establishing its role as a key positive regulator of salt stress tolerance [28]. In transgenic tobacco and lemon plants, the overexpression of *PtrERF109* of *Poncirus trifoliata* upregulates peroxidase-encoding genes, strengthening antioxidant capacity and enhancing cold tolerance through effective ROS scavenging [29]. Additionally, *ERF109* plays an important regulatory role in leaf response to iron (Fe) deficiency, which is closely linked to the regulation of pathogen defense and photosynthesis efficiency in plants [30]. These findings prompted the hypothesis regarding the functional role of *PdbERF109* in salt stress responses of *P. davidiana* × *P. bolleana*. To validate this, we stably transformed *PdbERF109*-overexpressing transgenic lines generated for functional characterization. Additionally, in high-salinity environments, maintaining moisture content is critical for plant survival, directly contributing to the maintenance of water balance, reduced salt ion transport, and mitigation of oxidative damage. Some of the literature points out that salt-tolerant plants undergo a series of physiological and biochemical reactions to maintain their water status, thereby limiting water loss and enhancing their salt tolerance [31]. This study demonstrated that the overexpression of *PdbERF109* enhances *P. davidiana* × *P. bolleana* water retention capacity under high salinity. In addition, plants manage water loss by regulating stomatal opening and closing, a mechanism that is central to their adaptation to environmental water conditions. In *Solanum lycopersicum*, *SlERF84* overexpression increased stomatal closure, preventing water loss and reducing overall plant water loss rate [32]. Based on these observations, we hypothesize that *PdbERF109* may reduce water loss in poplar by modulating stomatal closure. Therefore, future research should focus on elucidating the physiological and morphological mechanisms through which *PdbERF109* enhances water conservation capacity.

This study demonstrated that under high-salinity conditions, WT poplars exhibit accelerated wilting and exacerbated damage compared to *PdbERF109*-overexpressing transgenic lines. Furthermore, enzymatic analyses have shown that the transgenic lines display significantly elevated activities of SOD, POD, and CAT while concurrently exhibiting reduced levels of MDA and H_2_O_2_ relative to WT. There have been extensive reports on the involvement of ERF family members in ROS scavenging research. For example, in *alba* × *P. glandulosa*, transgenic overexpressing *ERF38* poplar exhibits enhanced salt tolerance, characterized by significantly elevated POD and SOD activities and lower H_2_O_2_ and MDA levels than the wild type [33]. Similarly, in *Nicotiana benthamiana*, the heterologous expression of pepper *CaERF2* improves salt tolerance through superior SOD, POD, and CAT activities and ROS scavenging gene induction [34]. Conversely, in *Betula platyphylla Sukaczev*, overexpressing *BpERF11* shows compromised salt tolerance with increased MDA, diminished SOD and POD activities, and ROS accumulation [35]. In summary, our study also shows that members of the poplar ERF family can participate in promoting antioxidant enzyme activity and ROS scavenging processes, indicating that the functions of ERF family members are relatively conserved across different species.

TFs regulate gene expression by specifically binding to cis-acting elements within the promoter regions of downstream genes, thereby activating or repressing transcription. This study demonstrated that *PdbERF109* specifically binds to the GCC-box and DRE motifs. The GCC-box is a key cis-acting element recognized by ERF TFs, primarily involved in plant defense and stress responses, particularly in the regulation of JA and ethylene signaling pathways. The DRE is a critical cis-acting element central to plant responses to abiotic stress, regulating the expression of stress-tolerance genes. For instance, in *Capsicum annuum*, *CaERF14* interacts with the GCC-box to regulate downstream genes, thereby mediating defense responses against salt stress [36]. In *O. sativa*, the tomato ERF protein TSRF1 enhances the expression of *MYB*, *MYC*, proline biosynthesis-related genes, and photosynthesis-related genes by binding to DRE and GCC-box elements, thereby consequently improving osmotic and drought tolerance [37]. In *Solanum lycopersicum*, *SlERF.B1* overexpression confers heightened sensitivity to salt stress in transgenic tomatoes, and *SlERF.B1* binds to the DRE in promoters to repress the expression of *SlARF5* and *SlER24* [38]. Similarly, *ThERF1*, an ERF gene from *Tamarix hispida*, negatively regulates salt tolerance by strongly suppressing the expression of *SOD* and *POD* genes, leading to reduced ROS scavenging capacity in *A. thaliana*, and *ThERF1* specifically binds both the GCC-box and DRE element [39]. In *A. thaliana*, ERF109 can combine with the GCC-box to encode two key enzymes in auxin biosynthesis, thereby affecting its growth and development [40]. Based on these findings, we thus hypothesize that *PdbERF109* may regulate poplar responses to salt stress by specifically binding to GCC-box and DRE elements within the promoters of downstream target genes.

In conclusion, our study demonstrates that *PdbERF109* functions as a key positive regulator of salt tolerance in *P. davidiana* × *P. bolleana* through multiple mechanisms, including enhancing ROS scavenging capability via elevated antioxidant enzyme activities, improving water retention capacity, and directly binding to GCC-box and DRE cis-elements to likely regulate stress-responsive genes. These functional attributes align with the conserved roles observed in *ERF109* homologs across diverse plant species, reinforcing the evolutionary significance of this transcription factor in abiotic stress adaptation. Our findings not only provide novel insights into the molecular basis of salt tolerance in woody plants but also position *PdbERF109* as a promising candidate gene for the molecular breeding of salt-resistant poplar varieties. While our study establishes the role of *PdbERF109* in enhancing salt tolerance, the specific downstream target genes directly regulated by *PdbERF109* through GCC-box or DRE binding have not been experimentally verified. Future studies may further reveal the precise transcriptional network controlled by *PdbERF109*. These investigations will significantly advance our understanding of how ERF transcription factors coordinate osmotic and ionic stress adaptation in woody plants.

## Figures and Tables

**Figure 1 plants-14-02800-f001:**
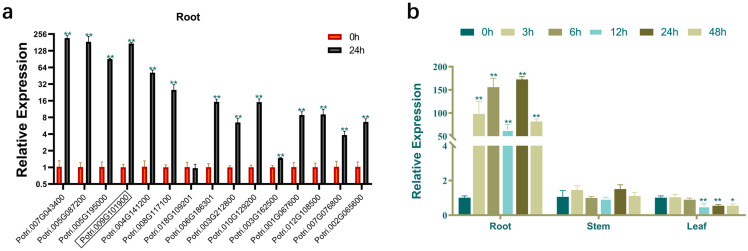
Expression pattern analysis of 15 ERF family members. (**a**) Fifteen *PdbERF* gene expression levels of root under salt stress at 0 h and 24 h. The boxed Potri.009G101900 is *PdbERF109*. The values on the y-axis have been processed using log_2_. (**b**) Tissue-differential expression patterns of *PdbERF109* in poplar under salt stress. The expression levels of 3, 6, 12, 24, and 48 h were calculated relative to 0 h. The error bars represent standard deviation. The asterisk indicates significant differences between the treatment group and the control group (*t*-test, * *p*-value < 0.05, ** *p*-value < 0.01).

**Figure 2 plants-14-02800-f002:**
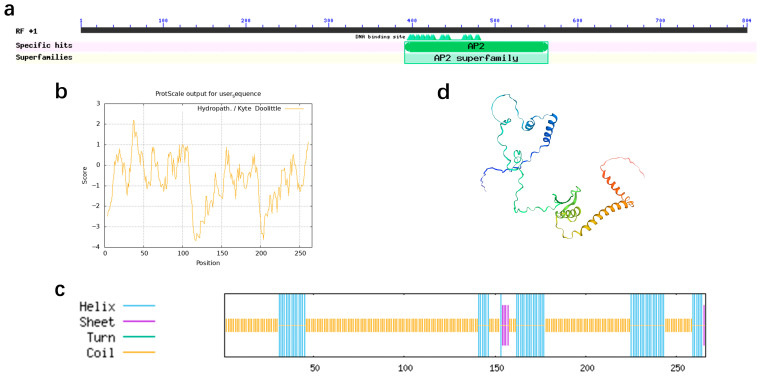
Protein characterization analysis of *PdbERF109*. (**a**) The conserved structural domain. (**b**) Hydrophilicity and hydrophobicity. (**c**) The secondary structure. (**d**) The tertiary structure.

**Figure 3 plants-14-02800-f003:**
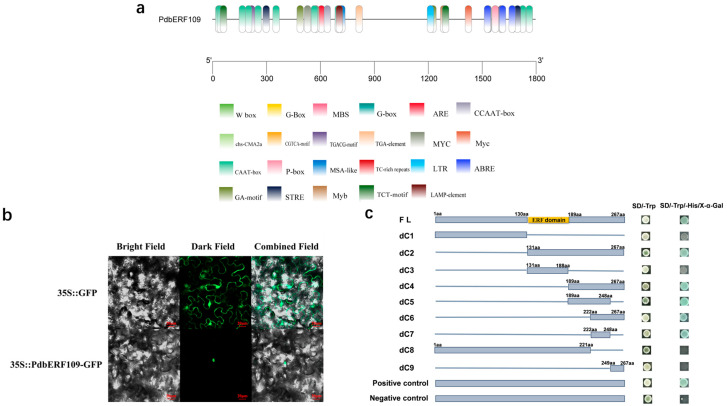
Biochemical characterization analysis of *PdbERF109*. (**a**) Stress-responsive cis-acting elements within the 1795 bp upstream region of *PdbERF109.* (**b**) Subcellular localization analysis of PdbERF109. The scale is 20 μm. (**c**) Transcriptional activation activity analysis of PdbERF109. Positive control: pGBKT7-53; negative control: pGBKT7. Amino acids are denoted as aa.

**Figure 4 plants-14-02800-f004:**
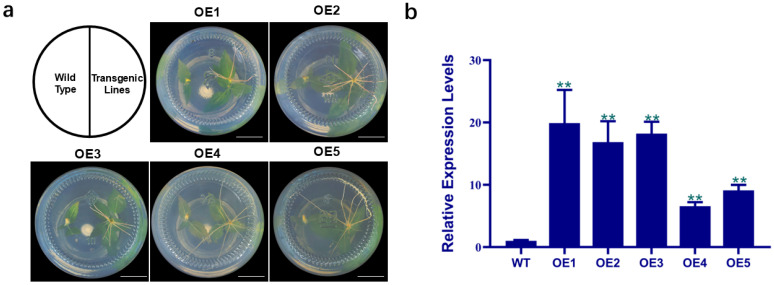
Molecular validation of *PdbERF109*-overexpressing transgenic poplar. (**a**) Transgenic poplars can root in the rooting medium containing 50 mg/L kan. The scale is 2 cm. (**b**) Analysis of the expression levels of *PdbERF109* in different transgenic lines. The expression level of *PdbERF109* was calculated relative to WT. The error bars represent standard deviation. The asterisk indicates significant differences (*t*-test, ** *p*-value < 0.01).

**Figure 5 plants-14-02800-f005:**
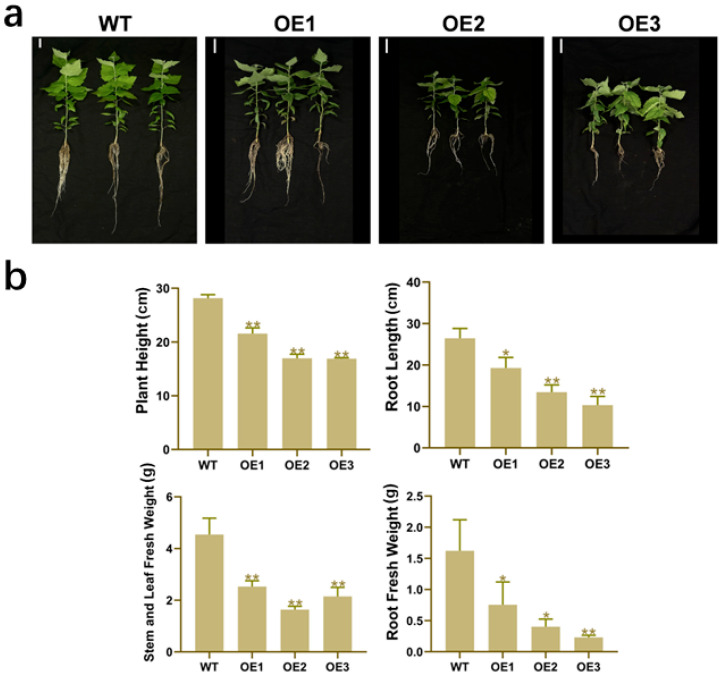
Phenotypic analysis of *PdbERF109*-overexpressing poplar. (**a**) Phenotypic comparison of WT and transgenic seedlings cultivated in soil. The scale is 4 cm. (**b**) Measurement of plant height, root length, and fresh weight. The error bars represent standard deviation (*t*-test, * *p*-value < 0.05, ** *p*-value < 0.01).

**Figure 6 plants-14-02800-f006:**
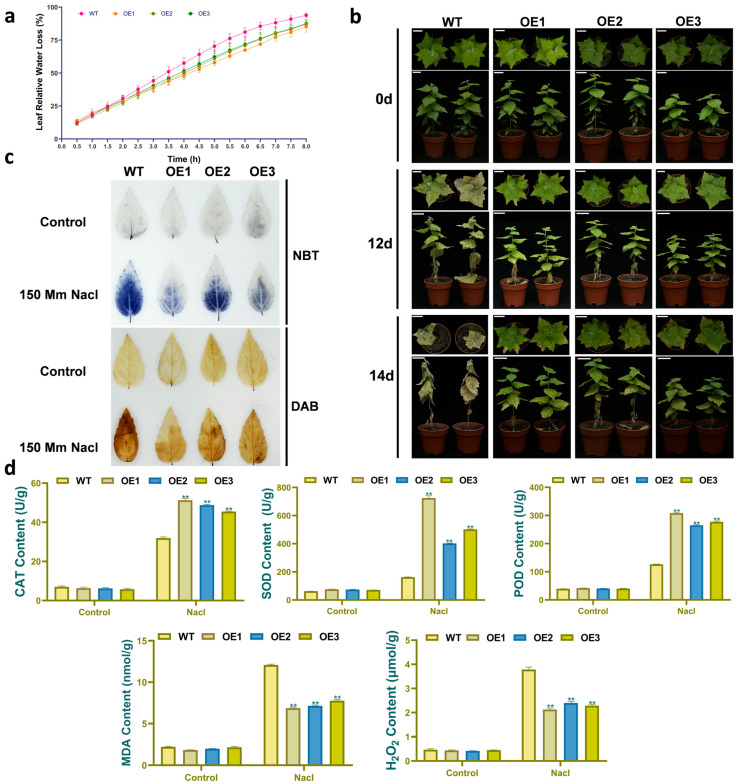
Phenotypic analysis of different transgenic poplars. (**a**) Leaf water loss rate. (**b**) Comparison of growth phenotypes among OE and WT poplar plants under salt stress. The scale is 4 cm. (**c**) Leaves of different lines treated with 150 mM NaCl concentrations for 2 h were stained with NBT and DAB; DAB reacts with H_2_O_2_ to produce a brownish-yellow precipitate; NBT reacts with O_2_^−^ forms to produce a blue precipitate. (**d**) H_2_O_2_ content, SOD activity, POD activity, CAT activity, and MDA content of different lines under salt treatment conditions; water treatment was the control. The experimental materials were leaves from soil-grown seedlings treated with water and salt for 50 days. The error bars represent standard deviation. The asterisk indicates significant differences (*t*-test, ** *p*-value < 0.01).

**Figure 7 plants-14-02800-f007:**
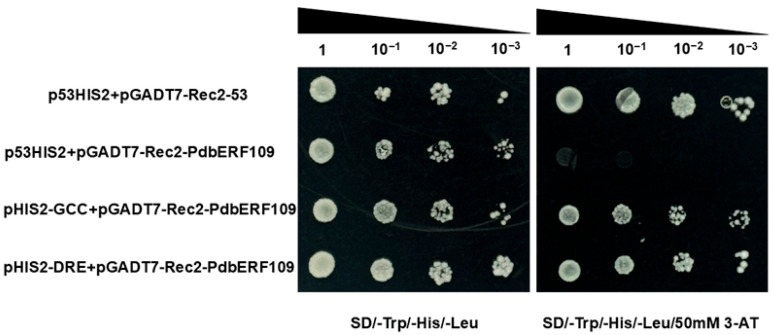
Analysis of *PdbERF109*-binding elements. p53HIS2+pGADT7-Rec2-53 was used as positive control; p53HIS2+pGADT7-Rec2-*PdbERF109* was used as negative control.

## Data Availability

Data is contained within the article.

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
