# Peer review of "Functional Analysis of PdbERF109 Gene Regulation of Salt Tolerance in Populus davidiana × P. bolleana"

_plants, 2025, doi:10.3390/plants14172800_

Round 1

Reviewer 1 Report

Comments and Suggestions for Authors

The authors present a well-structured research paper addressing agricultural challenges in northern China and elsewhere. The paper focuses on Poplar ERF genes, which appear to not have been studied in detail in their relation to salt tolerance. From a selection of 15 ERF genes, the authors identify 14 genes that are overexpressed in response to salt treatment. One of these genes, ERF109, is selected for further analysis by overexpression in plants. The paper investigates cellular localization, transcriptional activation domains and phenotypical changes.

The paper is interesting; however, I find that the current data relies too heavily on the observed overexpression of ERF109, missing controls implementing other overexpressed proteins. Acknowledgment of previous work on ERF109 in other plants is also lacking, particularly when binding to GCC-boxes has been shown for Arabidopsis ERF109.

In additional, I find that the paper in many cases lacks explanations of what the observed results mean. Starting results with a brief explanation of why and what is investigated and ending with a short remark on what was discovered may improve the flow of the paper.

Major comments

  1. Choice of ERF109 and comparison to other ERFs
  • Several genes are overexpressed to the same level as ERF109, prompting the question of why ERF109 was specifically selected and why other genes were not studied further. I am wondering if the subsequent observations are unique for ERF109 or whether other of the ERF genes would have similar phenotypes.
  1. Specificity of the transcriptional response
  • For the analysis of overexpression of ERF genes in response to salt stress, is this a specific response or is it simply a response to stress? The manuscript does not consider induction from a generic stress response.
  1. Overexpression interpretation and recommended control
  • Are the subsequent observations that overexpression of ERF109 improves salt tolerance specific? Could the observed effect simply be the result of overexpression influencing the plant stress response? In addition to comparison with WT, I would compare with a transgenic plant overexpressing something else such as an ERF gene that was not expressed in response to salt stress (fx: Potri.018G109201).
  1. Figures
  • Figure 1a: The y-axis is cut into three pieces making any visual comparison of the bars impossible and skews the scale. Consider using a logarithmic y-axis. The same axis issue also applies to panel b. The gene codes are not used elsewhere in the paper, one might guess the boxed gene is ERF109, but it is hard to know. Give readable names and put a table with gene names and codes in the supplementary.

Minor / Other comments

There is a considerable amount of double spaces, misplaced periods and similar throughout the paper. This should be addressed.

Some inconsistencies regarding the species, gene, protein name typesetting. Eg. line 123 “PdbEFF”, the species name should be in italics “PdbERF”. Personally, I would write species name in italics, protein name in normal letters and gene names in lower case italics.

Values are frequently given with a lot of significant digits probably not warranted by the experimental uncertainty. Reduce the number of significant digits so that the value is not given with higher precision than the associated error. Eg: lines 254, 263-266, 332-334.

Figure texts are quite lacking and do not describe what is shown in the figures.

The Yeast one-hybrid constructs are not well explained. What does the ‘p53’ and the ‘53’ mean? I’m assuming a p53 binding site and the p53 protein. Please add more detail here. The current controls do not really test the ERF109 specificity. The controls should probably include ERF109 but use a mutated DNA binding site.

The bioinformatic analysis appears more or less irrelevant to the rest of the paper (what is the point of giving the atomic composition of a protein?). You could replace it with a brief explanation of the protein (AP2/ERF DNA binding domain and disordered regions with some secondary structure propensity). Figure 2 includes two graphs with no visible data and none of it is ever mentioned again. This section should probably be removed unless the authors find a reason to keep it and improve the figure dramatically. A suggestion could be to perform a transcriptional activation domain prediction which would connect nicely to the later identification of that domain.

An activation domain in ERF is identified, but there is no discussion or subsequent mention of this. At the very least, the sequence of the minimum protein fragment should be shown. This should be compared with alignments of representative species ERF109 homologs (eg. Arabidopsis) and other ERF proteins. I would suggest doing replacing the copy-paste from Expasy ProtParam with a protein homology analysis. 

Line 308: the residues indicated do not exist in the protein. The authors seem to confuse amino acids and nucleotides. This is also the case in Figure 3c. This must be addressed; it is a substantive error that affects interpretation of the molecular mapping and activation domain results.

Line 340: The water loss rate is not a rate. Looking at the data, the lines appear to not go through zero, which would be expected. I would suggest modifying the analysis so that you are able to pick up the expected plateau when the leaves are close to completely dry. Try to find a suitable model so that you can obtain a rate constant.

Line 386: Three binding elements are mentioned, but only two are analyzed for ERF109 binding. Why is ABRE mentioned but not investigated? In addition, AtERF109 has been shown to bind GCC-boxes. What is the rule here? Do all AP2/ERF DBDs bind GCC and DRE (and ABRE) and do we expect ERF109 proteins from different species to have different DNA specificity?

Supplementary: The sequence of the PdbERF109 gene and protein would be of interest. I have two supplementary files: Table S1 and Table S2. There is no Figure S1. To my knowledge, Figure S1 is also not mentioned in the text.

References: The bioinformatic tools are not referenced and should be. It seems that important and relevant works on ERF109 are not referenced (eg. PMID: 29686365, https://doi.org/10.1038/ncomms6833, 10.1186/s12870-016-0908-z).

Specific short notes

  • Line 50: correct language (“There is report indicates that …”).
  • Figure 1a: Is it correctly understood that the red bar is always equal to 1? If so, it is redundant.
  • Figure 1b: color selection of bars highlights the 12 h data, why?
  • Line 169: SOMPA appears to be actually called SOPMA according to the url. There should be a proper reference in addition to url.
  • Figure 3: the figure text is mixed up between panels b and c. What is imaged in panel b?
  • Line 308: “aa” usually refers amino acids, which is not possible here.
  • Line 332: consider reporting as mean ± SD.
  • Figure 6c: shows colored leaves but there is no explanation of what the coloring means. There are some indications in the text, but it requires guessing.
  • Line 44, …, 386, 461: space between last letter and reference. Inconsistent. Appears seven times.
  • Line 467: floating reference [34]

Reviewer 2 Report

Comments and Suggestions for Authors
  1. Line70: why is Jatropha curcas important in this introduction?
  2. Line 79: please note what genus and species SmERF1 is from.
  3. Lines 100-118: slightly redundant with the preceding paragraph.
  4. Line 120: what is "environmental governance"?
  5. Line 135: lux is an inappropriate unit of measurement for plant research. The units of lux are not micromole photons m-2s-1.
  6. Line 138: what was the actual age/ size of the treated plants?
  7. Line 139: 150 mM NaCl = 0.3 osmolal = -0.72 MPa osmotic pressure. This is a moderate osmotic stress. In order to separate Na effects from osmotic effects, parallel experiments with KCl must be done. Presumably the treatment was a single application of NaCl was done as a soil drench. Were the plants in pots and the NaCl allowed to drain? What volume of NaCl was applied? How do the authors know what the actual NaCl concentration in the soil was?  What volume of soil were the plants in (i.e., pot size)?  Were the plants large compared to the soil, as a large plant in a small pot is not comparable to a small plant in a large pot.
  8. Line 141: I assume the samples were flash frozen in liquid N2 and stored at -80 C until use? Was the soil substrate was washed off the roots first?  How woody were the stems and were the whole stems collected?
  9. Lines 145-148: see previous comment on the need for a KCl control series.
  10. Line 152: citation for the CTAB method?
  11. Line 180: what protocol was used to grind the tissue? Were the roots and stems woody, required a different grinding method than the leaves?
  12. Line 182: did the CTAB method require modifications for the root and stem samples compared to the leaf samples? Again, a citation is necessary.
  13. Lines 185-187: it is now standard protocol to use two or more endogenous reference genes. Why only one? Why this one?  There are typically more than one actin gene in a genome. The actual actin gene ID was not listed in Supplementary Table 2.  Was its stability assessed over time?  Please note that citing another published study is generally insufficient unless the treatment regimens are identical.
  14. Lines 195-196: citation? Manufacturer for the confocal microscope?
  15. Lines 204-206: citation for the method?
  16. Lines 209-211: did the authors check for residual Agrobacterium infection by performing PCR with VirG primers?
  17. Lines 216-218: presumably during the salt and water treatments, but when - once the treatment was finished? How were the leaves made turgid? Floating on water for some period of time?
  18. Line 249: please confirm which tissue is referred to in Figure 1a. The figure legend should reflect this as well.
  19. Lines 253-254: fold or times? Please confirm, as they are mathematically distinct and can result in vastly different values.
  20. Figure 2: difficult to read even under magnification.
  21. Figure 3c: difficult to read.
  22. Figure 6, panel d: were these leaves and how long was the salt treatment?
  23. Line 382: “Nowadays” is a colloquial expression. Perhaps “During the past two decades…”
  24. Line 401: a point to consider: the in planta experiments constitute a salt shock as opposed to a typical, gradual increase in salt levels. The authors should consider how realistic their experiments were compared to salt levels in northern China.  Are the soils pretreated or leached with non-saline water prior to tree planting?
  25. Line 409: suggest "upregulated" rather than "changed". "Changed" is a neutral term that could also mean "downregulated" in this context.
  26. Lines 415-418: please confirm that Ptr109 was actually overexpressed in Poncirus trifoliata. The terminology used in ref 25 is confusing at best, but the title indicates that EF109 of trifoliate orange was investigated and over expressed. The authors should consider adding a phylogenetic tree to the manuscript (possibly in Supplementary Materials) with other woody species including Poncirus trifoliata.
  27. Line 432: how closely related to PdbERF109 is SlERF84?
  28. Line 436: is stomatal closure truly an example of a morphological alteration? A better example is the phenotypical differences between WT and OE 1, 2, and 3.
  29. Line 448: a sequence comparison between these ERFs might allow a hypothesis to be formed as to why some ERFs enhance salt tolerance/ ROS scavenging and others do not.
  30. Line 460: again, a sequence comparison might be useful.

Reviewer 3 Report

Comments and Suggestions for Authors

In this study, the authors quantified the expression levels of PdbERF genes in P. davidiana × P. bolleana before and after salt treatment, based on the gene structure of ERF family members in Populus trichocarpa. They then performed transgenic and functional validation of the salt stress–responsive gene PdbERF109 to assess its role in salt tolerance. The stated aim is to contribute to the theoretical understanding of molecular mechanisms underlying salt stress responses in poplar. The article is generally well written and clearly structured; however, I have several comments and suggestions that could help strengthen its scientific rigor and overall impact.

-The abstract should integrate key numerical results rather than relying solely on descriptive statements.

-The introduction is overly long and contains extensive background information without adequate citations.

-The aim of the study should be revised to be more precise and informative, clearly stating the specific research question, its significance and how the approach takes addresses the identified gap.

-The data analysis section requires improvement.

-Figure 2 requires enhancement in image resolution and overall quality to ensure quality and legibility.

-The discussion should provide deeper interpretation of the results, supported by critical comparison with relevant previously published studies.

- The authors should include a clear section discussing the limitations of the study.

Round 2

Reviewer 2 Report

Comments and Suggestions for Authors

The authors have adequately addressed the issues I raised.  

Reviewer 3 Report

Comments and Suggestions for Authors

I have carefully examined the revised version of the manuscript and the authors’ responses. I am satisfied that all the concerns raised in the previous round have been adequately addressed. The revisions have improved the clarity, quality, and scientific rigor of the work.